# Novel device for assisted vaginal birth: using integrated qualitative case study methodology to optimise Odon Device use within a feasibility study in a maternity unit in the Southwest of England

Emily J Hotton [1,2] Natalie S Blencowe [3] Nichola Bale,[2] Erik Lenguerrand,[4] Tim J Draycott,[2] Joanna F Crofts,[2] Julia Wade [5]

For numbered affiliations see end of article.

**Correspondence to**
Dr Emily J Hotton;
emily.hotton@nhs.net

## ABSTRACT

**Objective** When novel devices are used 'in human' for the first time, their optimal use is uncertain because clinicians only have experience from preclinical studies. This study aimed to investigate factors that might optimise use of the Odon Device for assisted vaginal birth.

**Design** We undertook qualitative case studies within the ASSIST Study, a feasibility study of the Odon Device. Each 'case' was defined as one use of the device and included at least one of the following: observation of the attempted assisted birth, and an interview with the obstetrician, midwife or woman. Data collection and thematic analysis ran iteratively and in parallel.

**Setting** Tertiary referral National Health Service maternity unit in the Southwest of England.

**Participants** Women requiring a clinically indicated assisted vaginal birth.

**Intervention** The Odon Device, an innovative device for assisted vaginal birth.

**Primary and secondary outcome measures** Determining the optimal device technique, device design and defining clinical parameters for use.

**Results** Thirty-nine cases involving an attempted Odon-assisted birth were included in this study, of which 19 resulted in a successful birth with the device. Factors that improved use included optimisation of device technique, device design and clinical parameters for use. Technique adaptations included: applying the device during, rather than between, contractions; having a flexible approach to the application angle; and deflating the air cuff sooner than originally proposed. Three design modifications were proposed involving the deflation button and sleeve. Although use of the device was found to be appropriate in all fetal positions, it was considered contraindicated when the fetal station was at the ischial spines.

**Conclusions** Case study methodology facilitated the acquisition of rapid insights into device function in clinical practice, providing key insights regarding use, design and key clinical parameters for success. This methodology should be considered whenever innovative devices are introduced into clinical practice.

## STRENGTHS AND LIMITATIONS OF THIS STUDY

⇒ Case study methodology including data from participant observation and/or interviews (with operators, midwives and/or women) was successfully used in an intrapartum setting to evaluate the use of a novel device, the Odon Device, for assisted vaginal birth (AVB).

⇒ Iterative data analysis and feedback of findings enabled rapid dissemination of findings to key stakeholders, and consensus regarding future alterations to device design, technique and selection criteria for optimal device use.

⇒ Observations were undertaken where possible; however, due to the unpredictable nature of AVBs it was not possible to attend them all, potentially impacting on the generalisability of our findings.

**Trial registration number** ISRCTN10203171.

## INTRODUCTION

Each year, approximately 82 000 women in the UK have an assisted vaginal birth (AVB).[1] In recent years, despite the known advantages of AVB, this rate has reduced with a corresponding increase in caesarean births in the second stage of labour. Current devices for AVB require a high level of training and skill (with additional expertise required to define the fetal position) and can be associated with significant maternal and neonatal morbidity if used incorrectly.[2] An innovative device that is easier and safer to use could increase women's access to AVB, which in turn would help to reduce the number of emergency caesarean sections performed in the second stage.[3 4]

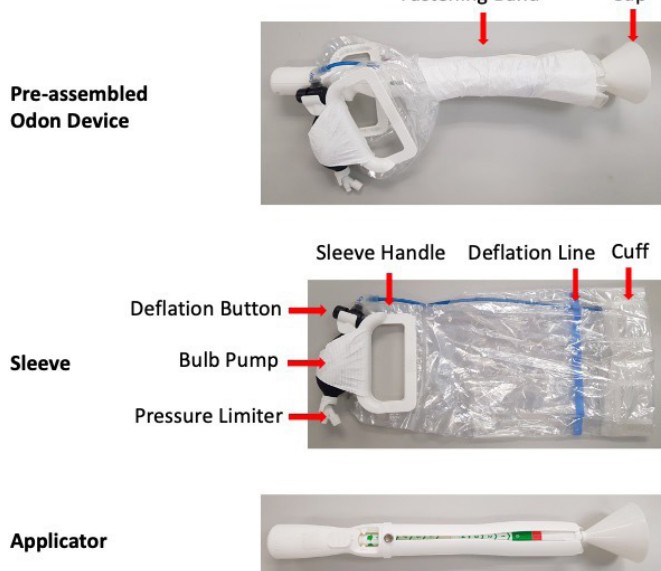

**Figure 1** Diagram of the Odon Device.

Before introducing devices into widespread practice, it is necessary to evaluate their safety and efficacy, and obtain CE marking. However, other factors—such as device technique, design and clinical parameters for use—are not routinely assessed, yet may ultimately limit their success. Preliminary feasibility work exploring these other factors may be valuable prior to evaluation within a definitive randomised controlled trial. The Odon Device (figure 1) has undergone rigorous preclinical studies,[5] simulation studies,[6 7] human factors[8] and phase I first-in-human investigation[9] which concluded that it appeared to be safe. However, the Odon Device has hitherto not been used in the intended population: women requiring an AVB. The Odon Device was originally designed by Jorge Odón and has since been developed by a team of clinicians and medical engineers. It assists birth using an inflatable air cuff attached to handles (figure 1).

This study applied qualitative case study methodology to examine in detail how the Odon Device (version 4.1) is used for AVB, and to determine what factors may impact on optimal use. The study was embedded in the ASSIST Study—a feasibility study of the Odon Device.[10 11]

## METHODS
### Research design
The ASSIST Study[10 11] was conducted in a maternity unit in the Southwest of England with full detail published elsewhere.[11 12] Integrated within the study was qualitative research using case study methodology to explore the factors that may influence optimum device use. Case study methodology is particularly suited to answering 'how and why' questions and providing in-depth contextual detail, essential in early evaluations[13] of complex interventions,[14–16] such as use of a novel device for AVB,[17] and has previously been used to explore surgical innovation.[18] In

this study, each 'case' was defined as one use of the device and included at least one of the following: observation of the attempted Odon-assisted birth and/or an interview with the obstetrician, midwife or woman. The researcher ensured that the use of the device in the study was compared against the Instructions For Use (IFU) document, which is mandated by regulatory bodies as one of the processes to ensure device safety and efficacy. Given the focus of this paper is of the technical aspects of device use, data presented reports observation and healthcare professional interview findings. Data reporting experiences of women are presented separately.[12]

### Participants
There were two groups of participants for the case studies: women and healthcare professionals (obstetricians and midwives). All women participating in the ASSIST Study were eligible to be included in the case study research and gave written consent.[12] All trained operators and midwives provided written consent. There were five operators, three consultants and two registrars.

### Sampling
Typical sampling for observation was in part purposive and in part opportunistic (ie, dependent on the researcher being on-site and available to conduct the observation). The aim was to include a range of clinical indications for AVB and a range of operators.

### Patient and public involvement
Patients and the public were involved in all aspects of the ASSIST Study, as previously reported.[10 12]

### Data collection
Included case studies comprised data from one or more of the following sources: observations of the AVBs and/or interviews with women, midwives and operators. Observations, including technical details, contextual factors and communication, were prospectively recorded on a bespoke observation schedule. Detailed observations of the operative steps performed by the obstetrician during AVBs were recorded enabling a stepwise account of the 'usual steps' to be generated and compared against the IFU. The original IFU documents were developed prior to the ASSIST Study during phase I clinical and simulation studies and included 22 operative steps.[8 9] In these IFU documents, the AVB was divided into six domains according to purpose (table 1). The IFU and instructional video for operators used for the Odon Device in the ASSIST Study can be viewed in online supplemental files 1 and 2.

All women who had the birth of their baby formally observed were invited to participate in an interview at day 1 postnatal and clinicians within 5 days following the assisted birth.[10] In line with usual practice in conducting case study research a flexible approach was taken to which data were collected for each case, based on the value of insights gained for each data source. Any method of data

**Table 1** Original components of application of the Odon Device for an assisted vaginal birth

| Component | Steps within component |
|---|---|
| Preparation | Checking clinical prerequisites for AVB. Lubricating the device. |
| Device application | Removing the fastening band. Applying the device onto a fetal head. |
| Cuff inflation | Ensuring the cuff is fully inflated in the correct position on the fetal head. |
| Applicator removal | Removing the applicator from the fetal head. |
| Traction | Following the J-shaped curve of the pelvis applying traction with contractions. |
| Removal of device | Deflating the air cuff as the fetal head is crowning. |

AVB, assisted vaginal birth.

collection (observation or interview) could be suspended if it was observed to be delivering no new insights.

### Data analysis
Data collection and analysis were iterative and ran in parallel using the six-step framework described for thematic analysis.[19] Data analysis was largely inductive although some deduction derived from using the IFU as a framework against which to evaluate what took place. All data for each case were read together to identify and organise codes. Codes were developed using text that captured significant views in the data, then grouped to reflect developing themes, with code descriptions and sample quotes assigned. Double coding of a proportion of interview transcripts (20%) was undertaken by JW. A narrative report was created for each case, triangulating all available data. Any issues requiring clarification were highlighted during the creation of the report, for exploration during subsequent interviews. Commonality and variances across cases were discussed between the researchers and used to further shape evolving themes and sampling. This systematic analysis supported rapid within-case and cross-case comparison. NVivo V.12 (QSR International, Melbourne, Australia) was used to organise data and support analysis.

### Feedback of findings
Iterative data collection and analysis enabled the rapid identification of key learning points or corrections to technique for dissemination to operators (see table 2). Key findings were relayed rapidly to operators using messages via an end-to-end encryption platform, regular face-to-face discussions and operator debriefs. Furthermore, following the 36th Odon-assisted birth, an interactive summit was held with key stakeholders (the clinical research team, design engineers, statisticians and funders), with the aim of sharing learning experiences and gaining consensus regarding any changes that may be suggested.

### RESULTS
Forty births were assisted with the Odon Device at North Bristol NHS Trust, UK, between October 2018 and January 2019. One case had no qualitative data because the researcher was unavailable, resulting in 39 case studies arising from 40 (97.5%) single uses of the Odon Device (table 2). Data for the case studies included 8 observations and accompanying interviews with the women, 19 midwife interviews, 37 operator interviews and 2 operator reflections (table 2). All births were assisted in the lithotomy position. Ninety per cent of women had a perineal tear, including 28 episiotomies, and three women (8%) sustained a third-degree perineal tear.[10] Nineteen births were successfully assisted with the Odon Device. Of those that were unsuccessful, 19 were assisted by forceps and two by caesarean section. There were no serious maternal or neonatal adverse events related to the use of the device and there were no serious adverse device effects. Four devices (10%) were ineffective due to a manufacturing fault.[10] Observations varied in length from 33 to 68 min. Interviews with women lasted 6.5–9.6 min, interviews with operators lasted between 5.4 and 26.1 min and interviews with midwives lasted 3.4–13.2 min. The shorter interviews with operators and midwives were all from cases in which the Odon Device was used successfully. Interviews for cases in which the Odon Device was unsuccessful were often longer as there were more aspects of device use to discuss. Another potential reason some interviews were short is that all operators and midwives were interviewed more than once, meaning they often did not have additional comments in subsequent interviews.

It became apparent that there were three factors contributing to optimisation of device use: (1) device technique, (2) device design, and (3) clinical parameters for device use (figure 2).

### Device technique
Suggested adaptions to the original IFU included (1) device application during rather than between contractions, (2) altering the application angle, and (3) deflating the air cuff as soon as any aspect of the blue deflation line became visible.

#### Device application with a contraction
The original IFU stated that the Odon Device should be applied between contractions, as was standard practice with forceps and ventouse. It became apparent during the first two attempted AVBs that this disimpacted the fetal head out of the pelvis and operators were unable to correctly place the device:

> Again, I had to use significant pressure to try and get the device over the fetal head. And loads of liquor came down during the application suggesting that there was some degree of disimpaction. (D1)

**Table 2** Summary of 40 cases investigating the Odon Device with adaptations made to device technique

| Case study No | Successful (S) and unsuccessful (U) AVB with Odon and mode of birth | Observation | Women | Interviews Operator | Midwife | Device issues |
|---|---|---|---|---|---|---|
| 1 | U—forceps | O1 | W1 | D1 | M1 | |
| 2 | U—forceps | | | D2 | M2 | |
| *Fundal pressure during device application tried* | | | | | | |
| 3 | S—Odon | | | D2 | M2 M3 | |
| *Deflation of the air cuff when only part of the blue line was seen introduced* | | | | | | |
| 4 | S—Odon | | | D2 | M4 | |
| 5 | U—forceps | | | D1 | M2 | |
| 6 | S—Odon | | | D2* | | |
| 7 | U—forceps | O2 | W2 | D1 | M5 | AD |
| *Accidental pressing of the deflation button first noted* | | | | | | |
| *Altered the angle of device insertion* | | | | | | |
| *Application during a maternal contraction introduced, use of fundal pressure removed* | | | | | | |
| 8 | S—Odon | O3 | W3 | D2 | M6 | AD |
| *Opened the sleeve handles during descent to monitor progression of fetal head first noted* | | | | | | |
| 9 | U—forceps | O4 | W4 | D1 | | |
| 10 | S—Odon | | | D2 | M7 | |
| 11 | S—Odon | O5 | W5 | D1 | M8 | AD |
| 12 | S—Odon | O6 | W6 | D1 | M9 | AD |
| 13 | S—Odon | O7 | W7 | D3 | M2 | |
| 14 | U—failed rotational forceps, emergency caesarean section | | | D1 | M7 | AD |
| 15 | U—forceps | | | D2 | M4 | |
| 16 | U—forceps | O8 | W8 | D2 | M10 | IBP |
| 17 | S—Odon | | | D1 | | |
| 18 | S—Odon | | | D3 | M11 | |
| 19 | S—Odon | | | D4 | | SST |
| 20 | U—rotational forceps | | | D4 | | IBP |
| 21 | U—emergency caesarean section | | | D3 | M6 | |
| 22 | U—forceps | | | D4 | | |
| 23 | U—forceps | | | D1 | M6 | IBP |
| 24 | S—Odon | | | D1 | | |
| 25 | U—forceps | | | D4 | | |
| 26 | U—forceps | | | D4 | | |
| 27 | S—Odon | | | D1 | | AD |
| 28 | U—forceps | | | D1 | | |
| 29 | S—Odon | | | D5 | | |
| 30 | U—forceps | | | D1 | | |
| 31 | U—forceps | | | D4 | | IBP |
| 32 | U—forceps | | | D1 | | |
| 33 | S—Odon | | | D5 | | |
| 34 | S—Odon | | | D2 | | |
| 35 | S—Odon | | | D2 | | |

Continued

| Case study No | Successful (S) and unsuccessful (U) AVB with Odon and mode of birth | Observation | Women | Interviews Operator | Midwife | Device issues |
|---|---|---|---|---|---|---|
| 36 | U—forceps | | | D4 | | |
| *Odon Summit held* | | | | | | |
| 37 | U—forceps | | | D1 | | |
| 38 | S—Odon | | | D4 | | |
| 39 | S—Odon | | | D4 | M6 | |
| 40 | U—forceps | | | D3 | | |

**Table 2** Continued

Bold italic steps denote key stages in the study that impacted on technique.
*Qualitative interview from obstetrician not obtained for this birth.
AD, accidental deflation; AVB, assisted vaginal birth; D, obstetrician; IBP, ineffective bulb pump; M, midwife; O, observation; SST, significant sleeve tear; W, woman.

By the third attempted birth, operators had adapted their technique to include fundal pressure to aid application, which resulted in successful device application and the first successful AVB. The use of fundal pressure, although successful, was not well tolerated by women without a regional anaesthetic:

Significant fundal pressure that was used at the time…she was uncomfortable…maybe that will be something up for review. (M3)

Following feedback from qualitative findings, the application technique was adapted again during the eighth birth. This was the first time the Odon Device was applied during a contraction without the use of fundal pressure, resulting in a successful application and birth. Fundal pressure was only used in a small number of births and quickly dropped from the technique as soon as application with a contraction was found to be successful:

I haven't used fundal pressure since delivery number two or three for me, but what has been very successful is putting it on during a contraction. I think. (D2)

### Device application angle

The original IFU stated that the device should be applied 'starting at 45° below the horizontal'. By the eighth attempted birth it was apparent that this was not optimal and operators naturally moved to a more 'horizontal' application:

I definitely pushed the device in at a much flatter angle, much more parallel with the bed than I had in the past… (D2)

All operators quickly agreed that the angle required might be dependent on factors such as fetal position and station:

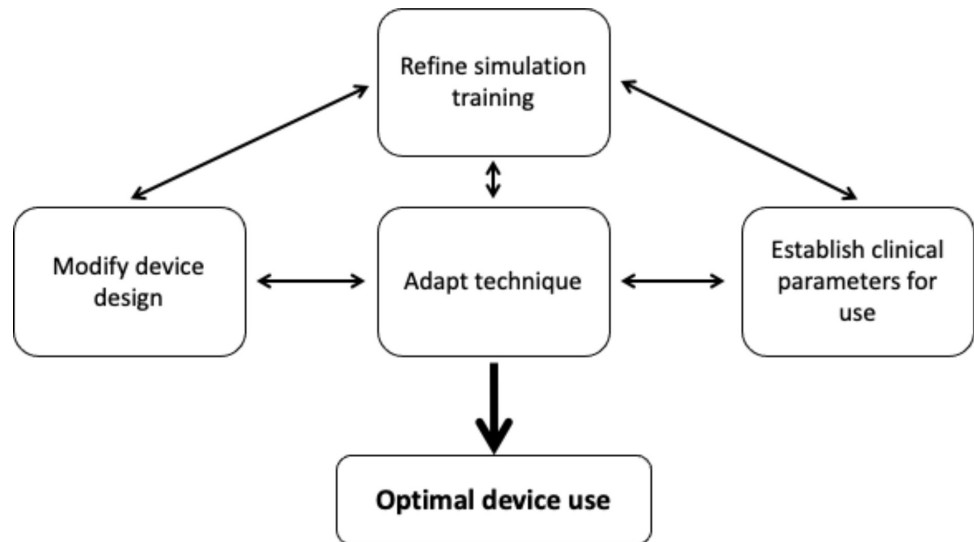

**Figure 2** How case study methodology may be able to determine optimal device use through bridging multiple factors relating to the device.

So I was kind of like, 'Oh, OP, it might be more, you know, it could be difficult because it's an OP…' (D1)

I think we've still got to continue experimenting or changing the angle of insertion. I think there may be an optimum angle of insertion or it may be that we have to change angle of insertion for different stations… (D2)

### Deflating the device

The original IFU stated that 'once you see the blue deflation line completely' the air cuff should be deflated. By the third attempted birth it became apparent to the observer that this was too late and that the optimum time for air cuff deflation seemed to be when any section of the blue line could be seen:

Noticed that it was not the anterior blue deflation line that the operator was looking at the deflate but the posterior deflation line, due to the fact that there is an acute J curve and anterior line not seen. Will need to change this in training. (O5)

These observations were fed back rapidly and iteratively to the Odon Device operators and further discussed with the wider research team at the Odon Summit (table 2).

### Device design and performance

Multiple potential device adaptations were noted during the case study research. Four design modifications for future device adaptations were identified: (1) strengthening the sleeve seal lines, (2) creating a wider opening between the sleeve handles, (3) altering the design of the deflation button, and (4) address the manufacturing fault that was identified.

### Sleeve seal lines and opening between sleeve handles

One operator noted that the sleeve seal lines tore during traction, on several occasions:

… the sleeve is not sturdy… it might actually rip it open, which has happened with me a few times. (D4)

During device inspection, it was noted that all devices had small tears (<2 cm) in the seal lines of the sleeve, and one had a significant tear (>7 cm). There was no evidence that any of these tears had had a negative effect on the function of the Odon Device, indeed the device with a significant tear achieved a successful Odon birth. Tearing was thought to have occurred when operators opened the sleeve handles before and between tractions to physically look at the station of the vertex. In contrast to standard devices used for AVB, there was little proprioceptive feedback to ascertain the station of the baby, so visual inspection was useful:

…I got the impression that the operator was unsure as to whether the head had descended so opened the handles to look inside the sleeve. (O5)

Following interviews, it was suggested that the opening between the two handles was made wider to enable operators to view the progression of the baby's head more easily. Ultrasound assessment was not used as this method was not routinely adopted in our unit at the time of the study.

### Deflation button

In six cases it was noticed that the operator accidentally pressed the deflation button. Each time this occurred, the cuff was reinflated immediately. All operators agreed that the design of the deflation button should be altered to reduce the risk of inadvertent activation (online supplemental figure S2):

Operator accidentally pressed the deflation button 'oh, whoops that was my fault, I'll just re-inflate'. (O3)

### Manufacturing fault

All devices were disinfected and inspected following their use as per protocol.[10 11] During this inspection, four devices were found to have an ineffective bulb pump which resulted in inadequate cuff inflation (table 2). Operators' comments during the attempted births reflected this, as the device did not act in the expected manner.

Yes, there was no grip… It just came out deflated, so it didn't feel right. (D4)

This prompted a rapid retrospective review of all used and stored devices to ensure that no other unsuccessful attempts were attributed to this fault, none were.

### Optimal clinical parameters for Odon Device use

The Odon Device was used to successfully assist births in all fetal positions. Midwives particularly noted how the device could help deliver a baby in the occipito-posterior position which is a technically more challenging position:

I think, probably, it could be quite universal as an instrumental device. It didn't seem to matter whether the baby was OA [occipito-anterior] or OP [occipito-posterior]… (M9)

However, although the device could be successful at assisting birth in all positions, it became apparent that for women with fetal station at spines or a more complex presentation (such as brow or nuchal arm) the device was not successful. Operators were either unsuccessful at applying the device correctly onto the fetal head or the device simply slipped off the fetal head with the initial traction:

So, it was direct OP at the spines, and it was almost coming to a brow, I could feel the orbital ridges…I was thinking, 'Oh, I'm really not sure that this is going to work.'…I didn't feel that was a failed Odon, that was a baby that was never going to come out vaginally (unsuccessful Odon-failed rotational forceps, emergency caesarean section). (D1)

As experience with the device increased, it became apparent that the device could be used comfortably

without a regional anaesthetic (with only perineal infiltration of local anaesthetic). Device use was noted to be better tolerated than bladder emptying by urethral catheterisation, a procedure that is less invasive:

> She actually found the catheterisation more uncomfortable than putting on the Odon Device with no analgesia at all. (D2)

### Feedback to operators

All qualitative case study findings relating to device technique, design and clinical parameters for use were presented to key stakeholders at the Odon Summit by the qualitative researcher. The case study research provided suggestions for device technique adaptation for some of the operative steps, but not for them all. It was agreed that there were still unanswered questions regarding the technique (such as which angle to use for application) and that further data were required to achieve this. Clinically important adaptations to device design were agreed on (including altering the deflation button design) and the clinical parameters for use were confirmed, with an agreement that the device should not be used if the vertex is at the level of the ischial spines.

### DISCUSSION

Case study research identified three areas that could optimise device use: (1) device technique, (2) device design, and (3) acceptable clinical parameters. Principal technique adaptations were centred on device application and deflation of the air cuff. The initial IFU specified a particular angle for device application; however, during clinical use it became apparent that this angle needed to be flexible and was less acute than originally specified; however, there was no consensus on the exact optimal angle and it was surmised that more data would be required to achieve this. Device modifications of altering the sleeve and deflation button were recommended for usability rather than to transform the functionality of the device. The manufacturing fault was quickly identified and rectified by the manufacturer through postuse device inspection. Optimal parameters for device use were proposed and focused primarily on the station of the baby, with use at station spines recommended to be prohibited. Adaptations to optimise device use were adopted by the manufacturer to create Odon Device (version 4.2) which was used in two further Odon Device feasibility studies, each studying 104 Odon-assisted births. These have recently closed to recruitment in the UK[20] and France[21] and aimed to address the unanswered aspects of optimal device use, specifically the technique. These findings will be published once follow-up and data analysis is complete. Case study research enabled systematic, rapid generation of data and understanding of device use that enabled the researchers and manufacturers to develop study protocols and device updates to support the ongoing investigation of the device.

### Strengths and limitations

This was the first time that research has been undertaken on the Odon Device in clinically indicated cases and indeed the first time case study research has been used to explore the use of devices for AVB. Device design and technique is unique to the device and although cannot be directly compared with other devices for AVB step by step, some comparisons and differences can be noted. The Odon Device, unlike other devices for AVB,[22] can only be successfully applied during a contraction or with maternal effort, even though techniques for traction once the device is applied appear similar. Clinical indications for use are slightly different from that of forceps and ventouse in the UK.[22] In the UK, all currently used devices for AVB are permitted to be used at station spines or below. We have demonstrated that this is not the case for the Odon Device, as we have demonstrated that this will not be successful. Interestingly, performing AVBs at station spines is not permitted in other countries.[23]

An AVB is a complex intervention, and this makes studying the use of the device challenging. Qualitative case study methodology has been used to explore technique in surgical procedures[18]; however, there are no published examples of case study methodology being used to investigate novel devices. The case studies integrated participant observation as well as interviews with operators, midwives and women to explore the introduction of an innovative device in context and in detail. The benefits of this were that experiences and views of all stakeholders were easily obtained, and we were able to investigate operator views in detail. Triangulation of data linked to a particular case led to insights for amendments for optimum device use being identified more rapidly than if a single source of qualitative data (eg, observation or interview only) had been used. Rapid dissemination of findings resulted in prompt adoption of beneficial techniques for use. By using this methodology and incorporating data from all stakeholders (operators, midwives and women) and observations we were able to gain a balanced and comprehensive assessment of the use of the device. When trying to understand optimal device use, operator interviews were found to be of crucial importance. Comparing case study data collected under different conditions (such as different analgesia, different presentations of babies, different operators) enabled commonalities and disparities in technique to be highlighted and thoroughly investigated. This enabled the clinical research team to propose evidence-based modifications to the device design and provide clarity on recommendations for clinical parameters for use. Case study methodology encouraged operators to reflect, critique and appraise their use of the device for each birth, resulting in enhanced and enriched communication between operators regarding their experiences through conversations and a dedicated operator messaging group. In future, data from encrypted social media platforms could be incorporated into the qualitative data for analysis. Reporting was undertaken following

the Standards for Reporting Qualitative Research[24] (online supplemental file 3).

There are limitations to this study. The aim of understanding the optimal operative steps for device use and thus confirming a finalised IFU was not met. For some operative steps consensus was reached as to the recommended course of action (such as applying the device with a contraction). However, for others more data were required (such as what specific angle of application to use). Case studies within the ASSIST Study were finite. Observations were undertaken where possible; however, due to the unpredictable nature of AVBs it was not possible to attend all assisted births. Indeed, none of the more complex attempted AVBs performed in the operating theatre were observed. This could have an impact on the generalisability of the findings as births undertaken in the operating theatre are often more technically challenging for operators. All interviews with clinicians were undertaken within 5 days following the assisted birth. Recollections of the clinicians may have been less accurate the longer the time between assisted birth and interview. The case studies were undertaken by a specialist trainee in obstetrics and gynaecology, meaning that preconceptions and existing knowledge may have influenced the collection and interpretation of the data, although at the time of commencing the case studies the researcher was naïve to the use of the Odon Device in the clinical setting. Lastly, operators may have changed their behaviours during observations, perhaps not reflecting their real-life practice.

There are two key next steps that should be considered. First, feasibility of the use of the Odon Device for AVB should be undertaken in different healthcare settings. Thus far, research has been undertaken in high-income settings where AVB is used regularly. Exploring device use in low-income and middle-income settings, where rates of AVB are lower than the UK and France, could help understand if there are further considerations for optimal device use that need to be addressed. Second, following the completion of the two further feasibility studies, a decision needs to be made as to whether the device is ready to be compared against available alternatives (forceps and ventouse) in a randomised controlled trial. As recommended by the Idea, Developement, Exploration, Assessment, Long-term follow up-Devices (IDEAL-D) collaboration,[25] researchers need to be satisfied that the technique, design and clinical parameters for use are sufficiently stable to enable this to happen.

## CONCLUSION
Case study methodology facilitated insights into optimal technique, design and clinical parameters for use of the Odon Device. Optimising use of a device is an essential prerequisite to evaluating outcomes, as it will impact directly on those outcomes and may result in lower than expected success rates. There were two clear factors that enhanced operator communication. First, systematic

triangulation of data from varying data sources provided a comprehensive, contextual overview of device use and rapid understanding of amendments required and, second, rapid feedback of insights as they emerged to operators. This also facilitated operator consensus building, which was key in understanding and developing the iterative adaptations to the device technique, design and clinical parameters for device use. This is of paramount importance for getting operator buy-in for the next steps of device evaluation. This methodology should be considered whenever innovative devices are introduced to clinical trials and settings. It allows for rapid assessment of device use and can support timely iterative adaptions to ensure there are minimal delays between device use in research and adoption in clinical practice.

**Author affiliations**
[1]Translational Health Sciences, University of Bristol, Bristol, UK
[2]Women's and Childrens Research, North Bristol NHS Trust, Bristol, UK
[3]Centre for Surgical Research, School of Social and Community Medicine, University of Bristol, Bristol, UK
[4]Musculoskeletal Research Unit, School of Clinical Sciences, University of Bristol, Bristol, UK
[5]Population Health Sciences, Bristol Medical School, University of Bristol, Bristol, UK

**Acknowledgements** The authors would like to thank all the women who agreed to take part in this research and all the maternity staff who enabled safe completion of the ASSIST Study. The research was affiliated with but not funded by the NIHR Biomedical Research Centre at University Hospitals Bristol and Weston NHS Foundation Trust and the University of Bristol.

**Contributors** EJH, NSB and JW developed the concept for the study. EJH performed all data collection and analysis with co-coding performed by JW. EJH wrote the initial draft of the manuscript, with support from NSB, JFC and JW. EJH, NSB, NB, EL, TJD, JFC and JW reviewed and approved the final manuscript. EJH is guarantor for this research.

**Funding** This work was supported by the Bill & Melinda Gates Foundation (INV-010180).

**Disclaimer** The views expressed are those of the author(s) and not necessarily those of the NIHR or the Department of Health and Social Care, and the MRC ConDuCT-II (Collaboration and innovation for Difficult and Complex randomised controlled Trials In Invasive procedures) Hub for Trials Methodology Research (MR/K025643/1).

**Competing interests** None declared.

**Patient and public involvement** Patients and/or the public were involved in the design, or conduct, or reporting, or dissemination plans of this research. Refer to the Methods section for further details.

**Patient consent for publication** Obtained.

**Ethics approval** This study involves human participants and was approved by the South Central–Berkshire REC, UK on 3 September 2018 (18/SC/0344), the MHRA on 9 August 2018 and the HRA on 3 September 2018. Participants gave informed consent to participate in the study before taking part.

**Provenance and peer review** Not commissioned; externally peer reviewed.

**Data availability statement** Data are available upon reasonable request.

**ORCID iDs**
Emily J Hotton http://orcid.org/0000-0002-8570-9136
Natalie S Blencowe http://orcid.org/0000-0002-6111-2175
Julia Wade http://orcid.org/0000-0001-6486-6477

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
