## [Reviewer comments · BMJ Open]

ARTICLE DETAILS

TITLE (PROVISIONAL)	Novel device for assisted vaginal birth: using integrated qualitative case study methodology to optimise Odon Device use within a feasibility study in a maternity unit in the Southwest of England
AUTHORS	Hotton, Emily; Blencowe, Natalie; Bale, Nichola; Lenguerrand, Erik; Draycott, Tim; Crofts, Joanna; Wade, Julia

VERSION 1 – REVIEW

REVIEWER	Feeley, Claire University of Central Lancashire
REVIEW RETURNED	07-Feb-2022

GENERAL COMMENTS	Thank you for giving me this opportunity to review this interesting, timely and novel device for assisted vaginal births. With few developments in this area, this is an exciting piece of work. I really enjoyed reading this well-crafted paper and robust piece of research. A massive congratulations to the research team. Comments below. Abstract- reads well, with clarity. Introduction – concise and clear, reads well. Methods – reads well, concise and clear. Minor point- would add a bit more about the PPI involvement, or at least add 'previously reported' to signpost this information. Data collection- what about the other cases? Need some explanation of the data collection around those and justify why only 8 are being discussed. Results – clear but it would be really useful to state what other maternal positions the Odon was used in and success rate. And perhaps consider that for further investigation? Women lying on their back can be a trauma trigger so anything that can avoid this or at least minimise the necessity of semi-recumbent is a great advance indeed. I am wondering about episiotomy or perineal trauma, I couldn't see anything about this but from watching the video it appears epis not necessary? Would suggest including some comments around this, for if the Odon reduces the risk of perineal trauma and/or epis then this vital information needs sharing. Findings- clear and easy to follow, reads really well. Minor point, referencing error under 'manufacturing fault'. Discussion and conclusion- reads really well, covers all points well with clarity. Key limitations and strengths were identified, explained and thought has been given to the next steps.
--

REVIEWER	Jiang, Hong Fudan University, School of Public Health
REVIEW RETURNED	26-Feb-2022

GENERAL COMMENTS	This study applied qualitative case study methodology to examine in detail how the Odon Device is used for AVB, and to determine what factors may impact on optimal use. The study is valuable in terms of delivering detailed information of the application of Odon Device for AVB. There are some questions regarding the Odon Devices and the study design.  1. Please have a brief introduction to Odon Device and its innovation compared with other devices for AVB. 2. Who invented the Odon Device? 3. What is the difference between case study and individual in-depth interview? 4. It was reported that some interviews last just for 3.4, 5.4 or 6.5 minutes. Would this be too short to obtain adequate information from study participants? 5. It seems no reflection of Odon Devices usage experience from women was reported. However, it is very important.
--

VERSION 1 – AUTHOR RESPONSE

REVIEWER 1, OVERALL COMMENT

Thank you for giving me this opportunity to review this interesting, timely and novel device for assisted vaginal births. With few developments in this area, this is an exciting piece of work. I really enjoyed reading this well-crafted paper and robust piece of research. A massive congratulations to the research team. Comments below.

Abstract- reads well, with clarity.

Introduction – concise and clear, reads well.

Methods – reads well, concise and clear.

REVIEWER 1, MINOR COMMENT

Would add a bit more about the PPI involvement, or at least add 'previously reported' to signpost this information.

RESPONSE

Thank you for this comment, we have more clearly signposted the reader to the earlier BMJ Open paper we published that has a very detailed description of the PPI.

EDITS

Page 7, 138-140.

Patients and the public were involved in all aspects of the ASSIST Study, as previously reported.^{10,12}

REVIEWER 1, DATA COLLECTION

What about the other cases? Need some explanation of the data collection around those and justify why only 8 are being discussed.

RESPONSE

Thank you for this comment and on review realise that we omitted to explain clearly where the data came from. We have amended the manuscript accordingly, removing results data from 'data collection' and providing further case information in 'results'.

EDITS

Data collection

Page 7, Line 143-144.

Data collection for eight of the cases included observation of the attempted Odon assisted birth. O Included case studies comprised data from one or more of the following sources: observations of the AVBs and/or interviews with women, midwives and operators.

Results

Page 8, Line 197-199.

Forty births were assisted with the Odon Device at North Bristol NHS Trust, UK, between October 2018 and January 2019. Qualitative data collection was not undertaken in one case because the researcher was unavailable, leaving 39 case studies (Table 2). Data for the cases studies included eight observations and accompanying interviews with the women, 19 midwife interviews, 37 operator interviews and two operator reflections (Table 2).

REVIEWER 1, RESULTS

Clear but it would be really useful to state what other maternal positions the Odon was used in and success rate. And perhaps consider that for further investigation? Women lying on their back can be a trauma trigger so anything that can avoid this or at least minimise the necessity of semi-recumbent is a great advance indeed.

I am wondering about episiotomy or perineal trauma, I couldn't see anything about this but from watching the video it appears epis not necessary? Would suggest including some comments around this, for if the Odon reduces the risk of perineal trauma and/or epis then this vital information needs sharing.

RESPONSE

Thank you for this comment. As for all other devices for AVB, the Odon Device was only used with women in the lithotomy position, as mandated in the Instructions for Use provided by the device manufacturer. This information has been previously published in Hotton EJ, Lenguerrand E, Alvarez M, O'Brien S, Draycott TJ, Crofts JF, et al. Outcomes of the novel Odon Device in indicated operative vaginal birth. *Am J Obstet Gynecol.* 2021;224(6):e607.

This is the same with regards to the use of episiotomy and perineal trauma. This manuscript purely focuses on the case study findings however our quantitative paper Hotton EJ, Lenguerrand E, Alvarez M, O'Brien S, Draycott TJ, Crofts JF, et al. Outcomes of the novel Odon Device in indicated operative vaginal birth. *Am J Obstet Gynecol.* 2021;224(6):e607 presents our episiotomy and perineal injury data. XX

EDITS

Page 9, Line 199

All births were assisted in the lithotomy position.

Page 9, Line 199-205

Ninety percent of women had a perineal tear, including 28 episiotomies and three women (8%) sustained a third-degree perineal tear.

There were no serious maternal or neonatal adverse events related to the use of the device and there were no serious adverse device effects. Four devices (10%) were ineffective due to a manufacturing fault.

REVIEWER 1, FINDINGS

Clear and easy to follow, reads really well. Minor point, referencing error under 'manufacturing fault'.

RESPONSE

Thank you for your kind comment. On our submitted version we are unable to see a referencing error under 'manufacturing fault' so have not made any edits to the manuscript, please can you clarify where this is so we can check again.

REVIEWER 1, DISCUSSION AND CONCLUSION

Reads really well, covers all points well with clarity. Key limitations and strengths were identified, explained and thought has been given to the next steps.

RESPONSE

Thank you very much for this positive comment.

REVIEWER 2, OVERALL COMMENT

A. This study applied qualitative case study methodology to examine in detail how the Odon Device is used for AVB, and to determine what factors may impact on optimal use. The study is valuable in terms of delivering detailed information of the application of Odon Device for AVB. There are some questions regarding the Odon Devices and the study design.

REVIEWER 2, COMMENT 1

Please have a brief introduction to Odon Device and its innovation compared with other devices for AVB.

RESPONSE

We thank you for this comment. These details have already been published in full in our manuscripts referenced in this paper but we have added a brief summary that incorporates this comment and comment 2 below. If the editors wish for more detail, we would be more than happy to provide further details if necessary.

EDITS

Page 5, Line 100-102.

The Odon Device was originally designed by Jorge Odón and has since been developed by a team of clinicians and medical engineers. It assists vaginal birth using an inflatable cuff attached to handles (Figure 1).

REVIEWER 2, COMMENT 2

Who invented the Odon Device?

RESPONSE

Please see response above to comment 1.

REVIEWER 2, COMMENT 3

What is the difference between case study and individual in-depth interview?

RESPONSE

Thank you for this comment, hopefully the edits that we have made in response to 'REVIEWER 1, DATA COLLECTION' has answered this. Case studies use data from one or more data sources, not necessarily just from interview. In our case, observations and/or interviews were used.

REVIEWER 2, COMMENT 4

It was reported that some interviews last just for 3.4, 5.4 or 6.5 minutes. Would this be too short to obtain adequate information from study participants?

RESPONSE

This is a very insightful comment and we have added information as to why some of the interviews are seemingly short.

EDITS

Page 9, Line 208-213

Interviews with women lasted 6.5 to 9.6 minutes, interviews with operators lasted between 5.4 and 26.1 minutes and interviews with midwives lasted 3.4 to 13.2 minutes. The shorter interviews with operators and midwives were all from cases in which the Odon Device was used successfully. Interviews for cases in which the Odon Device was unsuccessful were often longer as there were more aspects of device use to discuss. Another potential reason some interviews were short is that all operators and midwives were interviewed more than once, meaning they often did not have additional comments in subsequent interviews.

REVIEWER 2, COMMENT 5

It seems no reflection of Odon Devices usage experience from women was reported. However, it is very important.

RESPONSE

We thank you for this comment and we agree that experience from women is vital. These details have already been published in full in our earlier BMJ Open paper entitled 'The Odon Device to assist vaginal birth: Observations of and insights into women's experiences as narrated by health professionals and women'. We have fully referenced this publication throughout this manuscript. Due to the limited word count we felt that it was appropriate to purely reference these rather than reiterate the details of the device as the focus of this paper is more on device use. If the editors wish, we would be more than happy to repeat this detail in this manuscript.

We now hope that you will consider this revised manuscript for publication in BMJ Open and are more than happy to answer any further queries you may have.

VERSION 2 – REVIEW

REVIEWER	Jiang, Hong Fudan University, School of Public Health
REVIEW RETURNED	05-Jul-2022
GENERAL COMMENTS	I have no further comments.